# Survival following sublobar resection after neoadjuvant therapy for T1N1-2M0 lung cancer

Hayley Reddington[1☯], Isabel Emmerick[1☯], Javier Diaz Collante[1☯], Aniket Maini[2☯],
Raghav Kanzaria[2☯], Allison Crawford[2], William Phillips[1], Mark Maxfield[1], Karl Uy[1],
Feiran Lou[1¤]*

1 Department of Surgery, Division of Thoracic Surgery, University of Massachusetts Chan Medical School,
Worcester, Massachusetts, United States of America, 2 University of Massachusetts Chan Medical
School, Worcester, Massachusetts, United States of America

☯ These authors contributed equally to this work.
¤ Current address:University of Massachusetts Chan Medical School, Department of Thoracic Surgery,
Worcester, Massachusetts, United States of America
* Feiran.lou@umassmemorial.org

## Abstract

### Background

Lobectomy remains the standard surgical approach for resectable non-small cell lung cancer (NSCLC), but sublobar resection is increasingly considered, particularly for small, early-stage tumors. The role of sublobar resection after neoadjuvant therapy is unclear. We compared survival after sublobar resection versus lobectomy among patients with T1N1-2M0 NSCLC treated with neoadjuvant therapy.

### Methods

Using the National Cancer Database (2011–2021), we identified patients ≥18 years with clinical stage T1N1-2M0 NSCLC who received neoadjuvant chemotherapy, chemoradiation, or chemoimmunotherapy followed by resection within 8 months. Patients with pneumonectomy, metastases, positive margins were excluded. Patients were grouped by resection type: sublobar (wedge or segmentectomy) vs. lobectomy (including bilobectomy and extended). Survival was assessed via Kaplan–Meier analysis and multivariable Cox regression.

### Results

Of 2,257 patients, 148 (6.6%) underwent sublobar resection. Compared to lobectomy, sublobar resection was associated with fewer lymph nodes examined (24% vs. 54% with >10 nodes, p<0.001) and shorter hospital stays (3.9 vs. 5.3 days, p<0.001). Stage distribution was similar. One-year survival was comparable for both groups in stage II and III disease. At 5 years, survival was lower for stage III patients

**Data availability statement:** The data underlying this study were obtained from the National Cancer Database (NCDB) Participant User File (PUF), a program administered jointly by the American College of Surgeons and the American Cancer Society. The authors do not have the right to publicly share the NCDB dataset used in this study with anyone not on the Participant User File (PUF) agreement, per NCDB policy. The dataset may be requested from the NCDB via NCDB_PUF@facs.org after completing the PUF application process at https://www.facs.org/quality-programs/can-cer-programs/national-cancer-database/puf/. The authors confirm that they did not have any special access privileges that others would not have.

**Funding:** The author(s) received no specific funding for this work.

**Competing interests:** The authors have declared that no competing interests exist.

undergoing sublobar resection (48.6% vs. 59.9%, $p < 0.01$), with adjusted analysis confirming worse survival (HR = 1.41, $p = 0.013$).

## Conclusions

Among patients with T1N1-2M0 NSCLC treated with neoadjuvant therapy, sublobar resection was associated with similar short-term but inferior long-term survival compared to lobectomy. Lobectomy remains preferred for patients with adequate reserve, though sublobar resection may be appropriate for select patients.

## Introduction

Lobectomy has long been the gold standard surgical approach for resectable non-small cell lung cancer (NSCLC), providing optimal oncologic outcomes through complete anatomic resection with systematic lymph node dissection [1,2]. However, sublobar resections (segmentectomy and wedge resections) have gained increasing interest, particularly for patients with small, early-stage, node-negative tumors and those with limited pulmonary reserve or significant comorbidities [3,4]. The landmark CALGB 140503 trial demonstrated that sublobar resection can achieve compara-ble survival outcomes to lobectomy in patients with stage IA NSCLC ≤ 2 cm, funda-mentally changing the surgical paradigm for early-stage disease [5]. Nevertheless, the potential role of sublobar resections beyond the scope of early-stage disease remains unclear.

Concurrent with evolving surgical techniques, neoadjuvant therapy has emerged as an increasingly important component of multimodal treatment for locally advanced NSCLC [6,7]. The rationale for neoadjuvant treatment includes improving resectability by shrinking tumor bulk, addressing micrometastatic disease, and guiding adjuvant treatment planning [8,9]. Recent advancements in neoadjuvant strategies, including immunotherapeutic agents, tumor biomarker profiling, and improved patient selection criteria, have enhanced treatment responses and expanded the pool of patients who may benefit from surgical resection [10,11]. Neoadjuvant therapeutic advances have created new opportunities for surgical intervention while raising important questions about the optimal extent of resection following treatment response.

Despite significant advances in both surgical technique and neoadjuvant therapy, a critical gap exists in the literature regarding survival outcomes following sublo-bar resection in patients who receive neoadjuvant therapy for small, node-positive NSCLC [12]. While existing studies have predominantly focused on early-stage, node-negative disease or patients undergoing upfront surgical resection, there is min-imal data examining the oncologic appropriateness of sublobar resection in patients with node-positive disease following neoadjuvant therapy [12–14]. The relevance of this treatment niche is particularly important since neoadjuvant treatment may alter tumor biology and nodal status, potentially improving the safety of parenchymal spar-ing resections [15]. Current clinical guidelines remain unclear about when sublobar

resections are oncologically appropriate after neoadjuvant treatment in small, node-positive tumors, leaving patients and surgeons without evidence-based recommendations [16,17].

The purpose of this study is to evaluate overall survival outcomes following sublobar resection compared to lobectomy among patients with clinical stage T1N1-2M0 NSCLC who received neoadjuvant therapy. We hypothesize that, after adjusting for patient and tumor characteristics, survival following sublobar resection would be comparable to lobectomy in this population.

## Materials and methods

### Database and patient selection

The National Cancer Database (NCDB) was utilized to evaluate our hypothesis. The NCDB is a comprehensive registry of patients that captures approximately 70% of newly diagnosed cancer cases in the United States [18,19]. The NCDB was queried for adult patients (≥18 years old) between 2011–2021 with clinical stage T1N1-2M0 NSCLC. Inclusion criteria ensured that patients underwent neoadjuvant therapy (chemotherapy, chemoimmunotherapy, or chemoradiation) and had a subsequent surgical resection. Only patients who had surgical resection within 8 months of neoadjuvant therapy were included to avoid salvage operations being captured in the cohort. Patients who had a pneumonectomy, metastatic disease, positive surgical margins, or who were missing survival data were excluded. Patients with clinical stages other than II or III were also excluded. Patients were grouped by resection type: sublobar (wedge or segmentectomy) vs. lobectomy (including bilobectomy and extended). This study was reviewed by the University of Massachusetts Chan Medical School IRB and deemed exempt (IRB #00001773).

### Statistical analysis

Patient demographics and clinical variables included age, sex, race, Spanish/Hispanic origin, insurance information and Charlson-Deyo comorbidity index. Facility data included facility type, surrounding area population, and geographic location. Cancer-related variables included year of diagnosis, laterality, grade, number of regional lymph nodes examined, number of positive regional lymph nodes, days between diagnosis and diagnostic/staging procedure, tumor size, lymph vascular invasion (LVI), and clinical and pathological stage based on 8th edition of the American Joint Commission on Cancer's TNM lung cancer classification [20]. Treatment-related variables included time to treatment initiation, type of neoadjuvant and adjuvant therapy, days from diagnosis to definitive surgery, and operative approach.

The primary outcome was overall survival, defined as the time from diagnosis to death or last contact, assessed at one and five years. Analyses were stratified by clinical stage, and subgroup analysis compared wedge and segmentectomy outcomes separately. Secondary outcomes included length of stay, 30-day unplanned readmission, 30-day and 90-day mortality, and vital status at last follow up.

Baseline demographic and clinical characteristics were compared between the sublobar and lobar cohorts. Pearson's chi-square or Fisher's exact tests were used to compare differences on categorical variables, and Student's $t$ test was used for continuous variables, as appropriate. Overall survival was evaluated using the Kaplan-Meier method, with survival curves compared using the log-rank test. Multivariable analysis was performed with a Cox proportional hazards regression model using backward selection, adjusting for relevant demographic, clinical and treatment covariates. A two-sided $p$ value $< 0.05$ is considered statistically significant. All analysis were conducted using SAS (version 9.4, SAS Institute, Cary, NC).

## Results

A total of 2,257 patients with clinical stage T1N1-2M0 NSCLC who underwent resection following neoadjuvant therapy between 2011 and 2021 were identified. Of these, 148 patients (6.6%) underwent sublobar resection and 2,109 (93.4%)

underwent lobectomy. Among the sublobar cohort, 55 patients had wedge resections, 86 had segmentectomies, and 7 had unspecified sublobar resections. Patients undergoing sublobar resection were slightly older, though not statistically significant (65.2 vs 63.8 years, p = 0.06). Sex, race, insurance status, comorbidity burden, and facility type were similar between groups. Tumor size distribution differed, with a higher proportion of sublobar patients presenting with tumors ≤2 cm (62% vs 51%, p = 0.036). Stage distribution was comparable between the sublobar and lobar cohorts (stage II: 14% vs 13%; stage III: 84% vs 86%, p = 0.13) (Table 1).

Patients who underwent sublobar resection had significantly fewer lymph nodes examined (≥10 nodes: 24% vs 54%, p < 0.001) and a lower mean number of positive nodes identified (1.4 vs 3.1, p < 0.001). Sublobar resection was associated with a shorter mean hospital stay (3.9 vs 5.3 days, p < 0.001). Thirty-day readmission, 30-day mortality, and 90-day mortality were low and did not differ significantly between groups (Table 1).

### Short-term survival

One-year overall survival was similar between lobectomy and sublobar resection for both stage II and stage III disease (Fig 1; S1 and S2 Tables). Among stage II patients, 1-year survival was 92.7% for lobectomy, and 95.2% for the sublobar group (87.5% for wedge resection, and 100% for segmentectomy). For stage III patients, 1-year survival was 93.7% for lobectomy and 91.1% for sublobar (93.5% for wedge resection, and 91.4% for segmentectomy). Differences were not significant (Fig 1).

### Long-term survival

At 5 years, survival outcomes diverged (Fig 2; S3 and S4 Tables). For stage II disease, 5-year survival was 56.6% for lobectomy, compared with 40.5% for sublobar resections (35.0% for wedge resection and 48.9% for segmentectomy). However, the difference in survival did not reach statistical significance (p = 0.29). In contrast, for stage III disease, lobectomy conferred significantly superior survival compared with sublobar resection (59.9% vs 48.6%, p < 0.01). Both wedge and segmentectomy cohorts demonstrated inferior survival relative to lobectomy, with nearly identical survival estimates at 5 years (47.7% and 48.0%, respectively).

### Multivariable analysis

The long-term survival benefit of lobectomy in stage III disease was further evaluated using Cox proportional hazards regression modeling (Table 2). Sublobar resection was independently associated with increased risk of death at 5 years compared with lobectomy (HR = 1.41, 95% CI 1.08–1.85, p = 0.013). Other independent predictors of worse survival included increasing age and receipt of neoadjuvant chemoradiation compared to chemotherapy. Female sex and later year of diagnosis were protective in this subgroup.

## Discussion

Our analysis of patients with clinical stage T1N1-2M0 NSCLC undergoing resection following neoadjuvant therapy demonstrates that sublobar resection has comparable short-term survival to lobectomy, but inferior long-term survival, particularly among stage III patients. While one-year survival did not significantly differ by resection type for either stage II or stage III disease, five-year survival was significantly worse in the sublobar group for stage III disease. Despite lower overall 5-year survival compared with lobectomy, patients who underwent sublobar resection still achieved 48.6% 5-year survival after neoadjuvant therapy, highlighting the procedure's meaningful long-term potential in carefully selected, high-risk individuals. These findings suggest that while lobectomy remains the oncologic standard, sublobar resection can still yield a substantial chance of long-term survival for patients who are poor lobectomy candidates or achieve strong treatment response.

**Table 1. NCDB patients with surgical resection following neoadjuvant therapy, 2011-2021 (n = 2257).**

| | Total n (%) | Sublobar cohort n (%) | Lobectomy cohort n (%) | p-value |
|---|---|---|---|---|
| n | | 148 (6.6) | 2109 (93.4) | |
| Facility type | | | | 0.79 |
| Community | 82 (3.6) | 7 (4.7) | 75 (3.6) | |
| Comprehensive Community Cancer | 652 (29) | 40 (27) | 612 (29) | |
| Academic/Research Program | 1053 (47) | 73 (49) | 980 (47) | |
| Integrated Network Cancer Program | 452 (20) | 28 (19) | 424 (20) | |
| Location | | | | 0.30 |
| New England (CT, MA, ME, NH, RI, VT) | 252 (11) | 14 (9.4) | 238 (11) | |
| Middle Atlantic (NJ, NY, PA) | 494 (22) | 43 (29) | 451 (22) | |
| South Atlantic (DC, DE, FL, GA, MD, NC, SC, VA, WV) | 454 (20) | 32 (22) | 422 (20) | |
| East North Central (IL, IN, MI, OH, WI) | 386 (17) | 16 (11) | 370 (18) | |
| East South Central (AL, KY, MS, TN) | 110 (4.9) | 6 (4.1) | 104 (5.0) | |
| West North Central (IA, KS, MN, MO, ND, NE, SD) | 121 (5.4) | 6 (4.1) | 115 (5.5) | |
| West South Central (AR, LA, OK, TX) | 126 (5.6) | 8 (5.4) | 118 (5.6) | |
| Mountain (AZ, CO, ID, MT, NM, NV, UT, WY) | 66 (2.9) | 6 (4.1) | 60 (2.9) | |
| Pacific (AK, CA, HI, OR, WA) | 230 (10) | 17 (11) | 213 (10) | |
| Age at diagnosis, mean (std) | 63.9 (9.0) | 65.2 (8.4) | 63.8 (9.0) | 0.06 |
| Sex | | | | 0.89 |
| Male | 979 (43) | 65 (44) | 914 (43) | |
| Female | 1278 (57) | 83 (56) | 1195 (57) | |
| Race | | | | 0.10 |
| White | 1931 (86) | 130 (88) | 1801 (86) | |
| Black | 209 (9.3) | 9 (6.1) | 200 (9.5) | |
| Asian American | 3 (0.1) | 0 (0) | 3 (0.1) | |
| Native Hawaiian/Pacific Islander | 5 (0.2) | 2 (1.4) | 3 (0.1) | |
| Asian | 78 (3.5) | 6 (4.1) | 72 (3.4) | |
| Other | 17 (0.8) | 0 (0) | 17 (0.8) | |
| Spanish/Hispanic | 75 (3.3) | 5 (3.5) | 70 (3.4) | 0.99 |
| Primary Payer | | | | 0.69 |
| Not insured | 25 (1.1) | 2 (1.4) | 23 (1.1) | |
| Private | 959 (42) | 65 (44) | 894 (42) | |
| Medicaid | 133 (5.9) | 5 (3.4) | 128 (6.1) | |
| Medicare | 1075 (48) | 71 (48) | 1004 (48) | |
| Other government | 35 (1.6) | 2 (1.4) | 33 (1.6) | |
| Charlson-Deyo Score | | | | 0.15 |
| 0 | 1429 (63) | 85 (57) | 1344 (64) | |
| 1 | 551 (24) | 46 (31) | 505 (24) | |
| 2+ | 277 (12) | 17 (11) | 260 (12) | |
| Year of Diagnosis | | | | 0.20 |
| 2011 | 204 (9.0) | 14 (9.5) | 190 (9.0) | |
| 2012 | 173 (7.7) | 16 (11) | 157 (7.4) | |
| 2013 | 196 (8.7) | 17 (11) | 179 (8.5) | |
| 2014 | 228 (10) | 18 (12) | 210 (10) | |
| 2015 | 242 (11) | 13 (8.8) | 229 (11) | |
| 2016 | 245 (11) | 10 (6.8) | 235 (11) | |

*(Continued)*

**Table 1.** (Continued)

| | Total n (%) | Sublobar cohort n (%) | Lobectomy cohort n (%) | p-value |
|---|---|---|---|---|
| 2017 | 210 (9.3) | 9 (6.1) | 201 (9.5) | |
| 2018 | 194 (8.6) | 8 (5.4) | 186 (8.8) | |
| 2019 | 194 (8.6) | 13 (8.8) | 181 (8.6) | |
| 2020 | 169 (7.5) | 11 (7.4) | 158 (7.5) | |
| 2021 | 202 (9.0) | 19 (12.8) | 183 (8.7) | |
| Laterality | | | | 0.67 |
| Left | 691 (31) | 42 (28) | 649 (31) | |
| Right | 1558 (69) | 105 (71) | 1453 (69) | |
| Regional lymph nodes examined | | | | <0.0001 |
| 0 | 53 (2.4) | 12 (8.1) | 41 (1.9) | |
| 1-9 | 740 (33) | 75 (51) | 665 (32) | |
| ≥10 | 1165 (62) | 36 (24) | 1129 (54) | |
| Unknown | 299 (13) | 25 (17) | 274 (13) | |
| Regional lymph nodes positive, mean (std) | 3.0 (4.7) | 1.4 (1.7) | 3.1 (4.8) | <0.0001 |
| Days between diagnosis and diagnostic/staging procedure, mean (std) | 8.9 (16.6) | 9.4 (17.6) | 8.9 (16.6) | 0.78 |
| Clinical Stage | | | | 0.13 |
| Stage I | 16 (0.7) | 3 (2.0) | 13 (0.6) | |
| Stage II | 296 (13) | 21 (14) | 275 (13) | |
| Stage III | 1943 (86) | 124 (84) | 1819 (86) | |
| Tumor size | | | | 0.036 |
| 0-2 cm | 1148 (51) | 91 (62) | 1057 (51) | |
| >2 to 3 cm | 923 (41) | 44 (30) | 879 (43) | |
| >3 to 5 cm | 96 (4.3) | 8 (5.5) | 88 (4.3) | |
| >5 to 7 cm | 12 (0.5) | 0 (0) | 12 (0.6) | |
| >7 cm | 29 (1.3) | 3 (2.1) | 26 (1.3) | |
| Lymphovascular Invasion | | | | 0.67 |
| Not present/not identified | 1117 (49) | 78 (53) | 1039 (49) | |
| Present | 480 (21) | 31 (21) | 449 (21) | |
| Unknown/indeterminate | 660 (29) | 39 (26) | 621 (29) | |
| Treatment started, days from Dx, mean (std) | 35.8 (27.7) | 35.5 (27.5) | 35.8 (27.7) | 0.90 |
| Neoadjuvant Therapy | | | | |
| Chemotherapy only | 991 (44) | 75 (51) | 916 (43) | 0.09 |
| Chemoimmunotherapy | 146 (6.5) | 12 (8.1) | 134 (6.4) | 0.40 |
| Chemoradiation | 1120 (50) | 61 (41) | 1059 (50) | 0.034 |
| Adjuvant Therapy | | | | |
| Any adjuvant therapy | 837 (37) | 61 (41) | 776 (37) | 0.28 |
| Systemic (chemotherapy and/or immunotherapy) | 559 (25) | 36 (24) | 523 (25) | 0.90 |
| Radiation | 477 (21) | 36 (24) | 441 (21) | 0.33 |
| Days from diagnosis to definitive surgery, mean (std) | 142 (38.5) | 140.1 (43.5) | 142.1 (38.2) | 0.59 |
| Operative Approach | | | | 0.24 |
| Open | 913 (40) | 51 (43) | 862 (49) | |
| Video-Assisted Thoracoscopic | 575 (25) | 44 (37) | 531 (30) | |
| Robotic-Assisted Thoracoscopic | 407 (18) | 23 (19) | 384 (22) | |
| Missing/unknown | 362 (16) | | | |

*(Continued)*

**Table 1.** (Continued)

|  | Total n (%) | Sublobar cohort n (%) | Lobectomy cohort n (%) | p-value |
|---|---|---|---|---|
| Surgical Inpatient Length of Stay, mean (std) | 5.3 (5.9) | 3.9 (3.1) | 5.3 (6.0) | <0.0001 |
| 30-day unplanned readmission | 77 (3.4) | 3 (2.1) | 74 (3.6) | 0.48 |
| 30-Day Mortality | 28 (1.2) | 2 (1.4) | 26 (1.2) | 0.71 |
| 90-Day Mortality | 64 (2.8) | 3 (2.0) | 61 (2.9) | 0.80 |
| Number of months between diagnosis and last contact or death | 54.0 (35.0) | 48.7 (35.7) | 54.4 (35.0) | 0.053 |
| Vital Status (at time of last contact or death) |  |  |  | <0.0001 |
| Dead | 1053 (47) | 92 (62) | 961 (46) |  |
| Alive | 1204 (53) | 56 (38) | 1148 (54) |  |

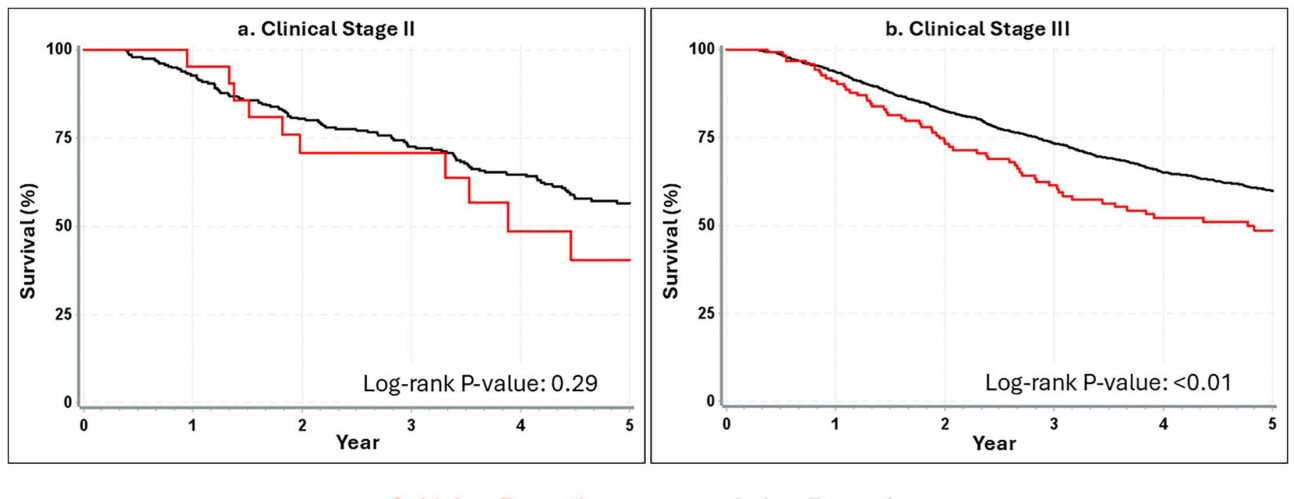

**Fig 1. Kaplan Meier curves for 1-year survival among clinical stage 2 (n = 296) and 3 (n = 1943) patients with T1N1-2M0 NSCLC.** (a) Kaplan–Meier survival curve for clinical stage II patients comparing sublobar and lobar resections. One-year survival was similar between the two surgical approaches, with no significant difference detected (Log-rank $P = 0.65$). (b) Kaplan–Meier survival curve for clinical stage III patients comparing sublobar and lobar resections. As with stage II disease, no significant difference in one-year survival was observed between surgical groups (Log-rank $P = 0.25$). Overall, short-term survival at one year did not differ significantly between sublobar and lobar resection for either stage.

## Short-term outcomes and perioperative considerations

Prior studies have highlighted low perioperative morbidity and mortality in parenchymal sparing resections [21]. It has also been demonstrated that sublobar resections may be associated with shorter hospital stays, reduced pulmonary compromise, and lower early complication rates compared to lobectomy [3,4]. Our data are congruent with this, as sublobar patients experienced shorter length of stay compared to the lobectomy cohort and equivalent short-term survival, suggesting that sublobar resection may be feasible and safe for selected patients following neoadjuvant therapy. Sublobar resection may provide early postoperative advantages compared to lobar resection. Importantly, the Charlson-Deyo comorbidity scores were similar between the groups, but other potential confounding variables are worth noting. Data such as pulmonary function, frailty, surgeon preference, and more granular information on comorbid conditions were not available for this analysis, which may impact outcomes.

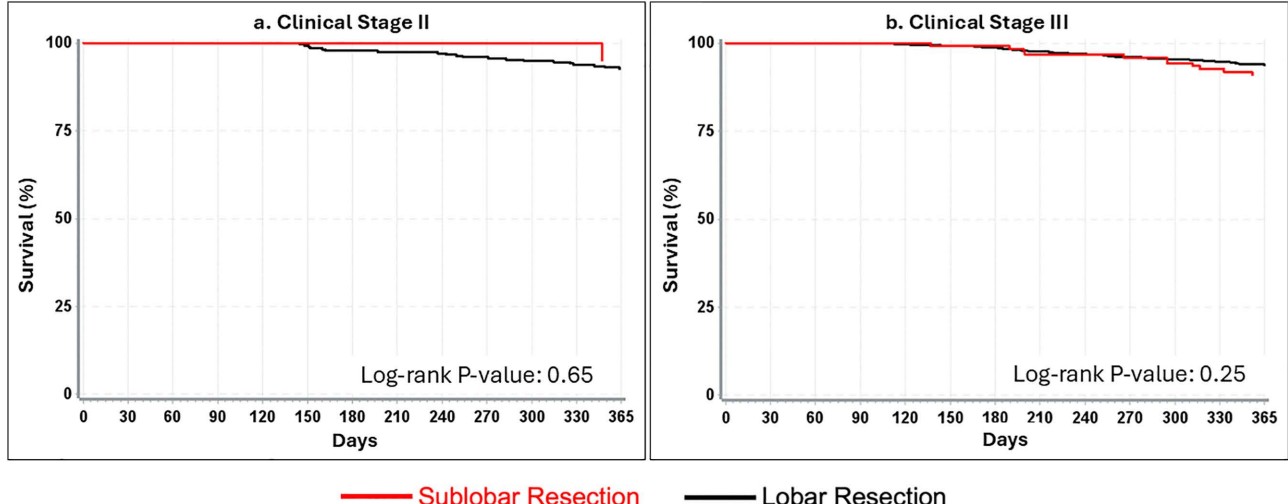

**Fig 2. Kaplan Meier curves for 5-year survival among clinical stage 2 (n = 296) and 3 (n = 1943) patients with T1N1-2M0 NSCLC.** (a) Kaplan–Meier survival curve for clinical stage II patients demonstrating that five-year survival did not significantly differ between sublobar and lobar resection (Log-rank *P* = 0.29). (b) Kaplan–Meier survival curve for clinical stage III patients showing a significant survival disadvantage associated with sublobar resection compared with lobectomy (Log-rank *P* < 0.01). Five-year survival was 49% for sublobar resection and 60% for lobectomy. These findings indicate that while long-term survival is similar for stage II disease, lobectomy provides a survival benefit for stage III disease.

**Table 2. Cox model for 5-year death among stage III patients (n = 1943).**

|  | HR | 95% CI | p-value |
|---|---|---|---|
| Sublobar resection | 1.41 | 1.08 – 1.85 | 0.013 |
| Age, per 10-year increase | 1.30 | 1.20 – 1.42 | <0.0001 |
| Female | 0.73 | 0.63 – 0.85 | <0.0001 |
| Year of diagnosis, versus 2011 |  |  | 0.009 |
| 2012 | 1.01 | 0.72 – 1.40 |  |
| 2013 | 1.00 | 0.73 – 1.37 |  |
| 2014 | 1.17 | 0.87 – 1.159 |  |
| 2015 | 0.81 | 0.59 – 1.11 |  |
| 2016 | 0.86 | 0.63 – 1.17 |  |
| 2017 | 0.69 | 0.49 – 0.97 |  |
| 2018 | 0.80 | 0.57 – 1.14 |  |
| 2019 | 0.63 | 0.43 – 0.91 |  |
| 2020 | 0.76 | 0.50 – 1.16 |  |
| 2021 | 0.66 | 0.40 – 1.08 |  |
| Neoadjuvant treatment, versus chemotherapy |  |  | <0.001 |
| Chemoimmunotherapy | 0.51 | 0.29 – 0.90 |  |
| Chemoradiation | 1.25 | 1.07 – 1.47 |  |

## Long-term outcomes and oncologic concerns

In contrast to the short-term findings, our study demonstrated inferior long-term survival associated with sublobar resection, specifically among patients with stage III disease. There are several key factors that may account for the diminished long-term survival in this cohort. First, patients who underwent sublobar resection had fewer lymph nodes examined

compared with lobectomy and a lower mean number of positive nodes detected. Inadequate nodal sampling could potentially lead to understaging, which may impact survival. Nodal assessment is a key quality indicator in surgical oncology that has long been recognized as critical to survival in NSCLC [2,22,23]. Extent of nodal evaluation may influence the inferior survival seen in our sublobar cohort, both by missing residual disease and by limiting adjuvant treatment decision-making.

Second, locoregional recurrence may contribute to inferior long-term survival in the sublobar cohort, inherent to the limited parenchymal resection by wedge and segmentectomy. Prior studies have demonstrated that achieving adequate parenchymal margins and systematic nodal dissection in early-stage NSCLC ensures oncologic equivalence of sublobar resection [5,13]. Therefore, in patients with node-positive disease, particularly after neoadjuvant therapy, limited resections may not provide sufficient local control to achieve long-term outcomes comparable to lobectomy. Unfortunately, recurrence data are unavailable in the NCDB, precluding direct assessment of recurrence patterns or disease-free survival. Disease-free survival and locoregional recurrence rate are arguably the most clinically relevant endpoints when comparing extent of resection, as they directly capture the adequacy of local tumor control. The inability to assess these outcomes represents a meaningful constraint of NCDB-based analyses in this setting, and future studies should prioritize these endpoints over overall survival alone.

Finally, although baseline characteristics such as age, comorbidity burden and insurance status were similar in both cohorts, selection bias may have influenced outcomes. Surgeons may preferentially offer sublobar resection to patients deemed less likely to tolerate a lobectomy. The NCDB does not include some factors that impact outcomes, such as pulmonary function testing and frailty. Although a multivariable model was constructed to decrease bias, not all survival related factors could be controlled. Patients who are high-risk due to pulmonary function or frailty making up the sublobar cohort may explain the survival inferiority.

### The role of sublobar resection

The utilization of parenchymal-sparing resections in NSCLC has evolved considerably in the past decade. For patients with small, node-negative tumors, the CALGB 140503 and JCOG0802/WJOG4607L randomized trials changed the surgical standard for early stage disease by demonstrating the oncologic non-inferiority of sublobar resection when compared to lobectomy [5,24]. However, these trials specifically excluded patients with nodal disease and did not address outcomes after neoadjuvant therapy.

These findings indicate that the use of sublobar resection should not be generalized to patients with more advanced disease. Other studies have shown that patients with locally advanced NSCLC undergoing induction therapy followed by surgery may benefit from more extensive resections, demonstrating that lobectomy remains superior when N2 disease is discovered at resection [12,14]. Our analysis adds to this growing body of literature suggesting that lobectomy should remain the oncologic standard for node-positive NSCLC, even in the context of neoadjuvant therapy. All things considered, the sublobar group demonstrated unexpectedly favorable long-term outcomes: one-year survival above 90% and 5-year survival nearing 50%, rates that historically paralleled lobectomy cohorts prior to advancements in neoadjuvant treatment [9,12]. These findings supports the feasibility and potential oncologic durability of sublobar resection in highly selected patients following neoadjuvant therapy. The adoption of neoadjuvant chemoimmunotherapy has introduced new complexities into surgical decision-making. Clinical trials such as NADIM and CheckMate-816 demonstrated improved pathological response and survival with neoadjuvant immunotherapy, with major pathological response rates exceeding 40% [11,25]. The results of these trials raise questions of whether more limited resection is sufficient in patients who have achieved near-complete pathological response. However, our findings suggest caution. Despite advances in systemic therapy, residual microscopic disease and inadequate nodal clearance remain significant risks in patients undergoing sublobar resection. Until prospective data demonstrate oncologic equivalence, lobectomy should remain the standard for patients with node-positive disease, regardless of neoadjuvant response. The NCDB does not capture pathological

complete response (pCR) as a structured data element, precluding response-stratified survival analysis. While pathological stage ypT0N0 serves as a proxy, missing pathological staging data and small subgroup sizes rendered this analysis infeasible. The chemoimmunotherapy subgroup was similarly underpowered, with only 12 sublobar patients receiving this regimen. Future studies should assess outcomes not only by surgical extent, but also by pathological response to therapy, as this may ultimately guide individualized surgical decision-making.

The long inclusion period of this study (2011–2021) encompasses substantial evolution in neoadjuvant treatment paradigms, from platinum-based chemotherapy to the integration of immune checkpoint inhibitors following landmark trials including CheckMate-816 and NADIM [11,25]. Our multivariable Cox model accounts for this temporal heterogeneity through adjustment for year of diagnosis, which was independently associated with improved survival in later years (p = 0.009), consistent with the impact of improved systemic therapies on outcomes over time. However, this covariate cannot fully capture the shift toward chemoimmunotherapy-based regimens after approximately 2018, nor the differential response rates and pathological downstaging associated with these regimens. Analyses restricted to the chemoimmunotherapy subgroup were not feasible in our dataset, as only 146 patients (6.5%) received this regimen, of whom only 12 underwent sublobar resection, precluding any meaningful survival comparison within this subgroup.

## Limitations

Limitations include a retrospective design that introduces inherent selection bias, unmeasured confounding, and missing data. The NCDB specifically lacks information on recurrence, disease-free survival, pulmonary function, performance status, and specific chemotherapy or immunotherapy regimens; variables that may influence procedure selection and long-term outcomes. Critically, pCR and major pathological response (MPR) are not captured as structured fields in the NCDB, as these require granular interpretation of narrative pathology reports outside the scope of registry abstraction. This precludes stratification of outcomes by treatment response, the variable most likely to identify patients for whom sublobar resection may be oncologically appropriate and represents a key limitation in the context of the evolving chemoimmunotherapy landscape. While pathological stage was available for a subset of patients, missing pathological staging data in approximately 45% of stage III patients precluded reliable comparative analysis of upstaging and downstaging rates between surgical groups. Furthermore, the chemoimmunotherapy subgroup was markedly underpowered for subgroup survival analysis, with only 12 sublobar patients receiving this regimen. The relatively small sublobar cohort overall (6.6% of the total sample) limits statistical power for subgroup analyses, particularly for wedge versus segmentectomy comparisons. Finally, the NCDB does not capture locoregional recurrence, which is arguably the most relevant endpoint for comparing extent of resection and a critical gap for future research to address.

## Future directions

Future research should prioritize study designs that capture outcomes not possible within administrative databases. Specifically, studies should incorporate pathological response assessment, including pCR and MPR, as primary stratification variables, alongside disease-free survival and locoregional recurrence as co-primary endpoints. Whether patients achieving pCR or MPR following neoadjuvant chemoimmunotherapy represent a subgroup in whom sublobar resection provides equivalent oncologic control with preserved pulmonary function is an urgent and clinically meaningful question that cannot be answered with existing registry data. Multi-institutional retrospective cohort studies leveraging institutional tumor registries, where granular pathology data are accessible, represent a feasible near-term approach. Patient selection criteria should integrate functional outcomes, pulmonary reserve, and quality of life alongside oncologic endpoints, given that preservation of pulmonary function remains a primary rationale for sublobar resection.

Finally, more work is needed to understand how immunotherapy and biomarker-driven therapies interact with surgical extent and long-term results as they continue to change the neoadjuvant landscape. Until such data is available,

lobectomy should remain the default surgical approach for operable patients with node-positive NSCLC after neoadjuvant therapy, with sublobar resection reserved for cases where lobectomy is not feasible.

## Supporting information

**S1 Table. One-year survival among clinical stage II patients (n = 296).** Kaplan–Meier curves demonstrate 1-year over-all survival for patients with clinical stage II NSCLC (n = 296) treated with lobectomy or sublobar resection (overall, wedge, or anatomic segmentectomy). At baseline, 275 patients were at risk in the lobectomy cohort and 21 in the sublobar group. One-year survival was 92.73% for lobectomy, 95.24% for the sublobar overall group, 87.5% for wedge resections, and 100% for segmentectomy. Standard errors are provided at each time interval to reflect precision of survival estimates. (DOCX)

**S2 Table. One-year survival among clinical stage III patients (n = 1943).** Kaplan–Meier curves show 1-year overall survival for clinical stage III NSCLC patients (n = 1,943). At baseline, numbers at risk included 1,819 for lobectomy and 124 for sublobar resections. One-year survival was 93.74% for lobectomy compared with 91.08% in the sublobar overall cohort. Among sublobar techniques, wedge resection demonstrated 93.5% 1-year survival and segmentectomy 91.4%. Standard errors are reported at each interval to quantify uncertainty. (DOCX)

**S3 Table. Five-year survival among clinical stage II patients (n = 296).** Kaplan–Meier curves depict 5-year overall survival for clinical stage II NSCLC patients (n = 296). At baseline, 275 patients were at risk in the lobectomy group and 21 in the sublobar group. Five-year survival was 56.56% following lobectomy and 40.48% for the sublobar overall group. Subgroup survival estimates at 5 years were 35.0% for wedge resection and 48.9% for anatomic segmentectomy. Standard errors accompany all estimates to indicate statistical precision. (DOCX)

**S4 Table. Five-year survival among clinical stage III patients (n = 1943).** Kaplan–Meier curves illustrate 5-year overall survival in clinical stage III NSCLC patients (n = 1,943) treated with lobectomy or sublobar resection. At baseline, numbers at risk were 1,819 for lobectomy and 124 for sublobar procedures. Five-year survival was 59.85% for lobectomy compared with 48.62% for sublobar resection overall. For sublobar subtypes, 5-year survival was 48.0% after wedge resection and 47.7% after segmentectomy. Standard errors are shown for all survival points. (DOCX)

## Acknowledgments

We thank the Department of Surgery and Division of Thoracic Surgery at UMass Chan Medical School for their support, the National Cancer Database for data access, and our colleagues for guidance. We also acknowledge the clinical teams and patients whose care informed this study.

## Author contributions

**Conceptualization:** Hayley Reddington, Aniket Maini, Raghav Kanzaria, William Phillips, Mark Maxfield, Karl Uy, Feiran Lou.

**Data curation:** Isabel Emmerick, Javier Diaz Collante, Allison Crawford.

**Formal analysis:** Isabel Emmerick, Allison Crawford.

**Investigation:** Isabel Emmerick, Mark Maxfield, Feiran Lou.

**Methodology:** Isabel Emmerick, Aniket Maini, Mark Maxfield, Karl Uy, Feiran Lou.

**Project administration:** Feiran Lou.

**Resources:** Karl Uy.

**Supervision:** Feiran Lou.

**Validation:** Allison Crawford, Feiran Lou.

**Writing – original draft:** Hayley Reddington, Javier Diaz Collante, Raghav Kanzaria.

**Writing – review & editing:** Hayley Reddington, Isabel Emmerick, Javier Diaz Collante, Aniket Maini, Raghav Kanzaria, William Phillips, Mark Maxfield, Karl Uy, Feiran Lou.

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
