## [Decision Letter · Decision Letter 0]

13 Mar 2026

PONE-D-26-01032Survival Following Sublobar Resection After Neoadjuvant Therapy for T1N1-2M0 Lung CancerPLOS One

Dear Dr. Lou,

Thank you for submitting your manuscript to PLOS ONE. After careful consideration, we feel that it has merit but does not fully meet PLOS ONE’s publication criteria as it currently stands. Therefore, we invite you to submit a revised version of the manuscript that addresses the points raised during the review process.

We look forward to receiving your revised manuscript.

Kind regards,

Luca Bertolaccini, M.D., Ph.D.

Academic Editor

PLOS One

Journal Requirements:

Reviewers' comments:

Reviewer's Responses to Questions

**Comments to the Author**

1. Is the manuscript technically sound, and do the data support the conclusions?

Reviewer #1: Yes

Reviewer #2: Yes

2. Has the statistical analysis been performed appropriately and rigorously? 

Reviewer #1: Yes

Reviewer #2: Yes

3. Have the authors made all data underlying the findings in their manuscript fully available?

Reviewer #1: No

Reviewer #2: Yes

4. Is the manuscript presented in an intelligible fashion and written in standard English?

Reviewer #1: Yes

Reviewer #2: Yes

5. Review Comments to the Author

Reviewer #1: The authors present an interesting and timely NCDB-based study evaluating outcomes of patients with clinical T1N1–2M0 NSCLC treated with neoadjuvant therapy and subsequent resection. The topic is clinically relevant and well aligned with current treatment paradigms.

I have a few comments and suggestions:

- Given the increasing adoption of neoadjuvant chemo-immunotherapy, a subgroup analysis focusing on patients treated with this strategy would be of particular interest, if feasible, even acknowledging the relatively limited sample size.

- Did the authors evaluate patterns of pathological upstaging or downstaging following neoadjuvant therapy? An exploratory analysis of patients experiencing stage migration could provide additional insight into treatment response and surgical decision-making.

- While recognizing the limitations of the NCDB regarding recurrence data, further clarification on the lack of disease-free survival and recurrence information should be emphasized, as these outcomes would be particularly relevant when comparing sublobar resection and lobectomy in this setting.

Reviewer #2: This manuscript addresses a clinically relevant question regarding the oncologic outcomes of sublobar resection after neoadjuvant therapy in patients with T1N1–2M0 NSCLC. The analysis is well conducted and clearly presented; however, some important aspects warrant further clarification.

Major Comment

An important element that appears to be missing from the analysis is the impact of pathological response to neoadjuvant therapy on survival outcomes. The neoadjuvant treatments included are highly heterogeneous (chemotherapy, chemoradiation, and chemo-immunotherapy) over a long study period. In the contemporary setting, neoadjuvant chemo-immunotherapy is associated with higher rates of major and complete pathological response, which are known to have strong prognostic value.

It would therefore be relevant to assess whether outcomes differ according to pathological response (e.g. pCR or nodal downstaging), particularly within the sublobar resection cohort. If feasible within the NCDB, an analysis stratified by pathological response would significantly strengthen the manuscript. If not feasible, this limitation and its potential implications should be more explicitly discussed, acknowledging the prognostic role of pathological complete response and its possible interaction with surgical extent.

Minor Comment

Given the long inclusion period (2011–2021) and the evolving neoadjuvant landscape, the potential impact of temporal changes in systemic therapy on outcomes could be further emphasized in the Discussion.

6. PLOS authors have the option to publish the peer review history of their article (what does this mean?). If published, this will include your full peer review and any attached files.

Reviewer #1: **Yes:** Pietro Bertoglio

Reviewer #2: No

---

## [Author Response · Author response to Decision Letter 1]

21 Apr 2026

Please note that the comments below are in a response letter labeled as "Response to Reviewers" and uploaded separately via Editorial Manager.

Response to Reviewers

"Survival Following Sublobar Resection After Neoadjuvant Therapy for T1N1–2M0 Lung Cancer"

Dear Editors,

We are grateful to the editors and reviewers for their thoughtful and constructive comments on our manuscript. We have carefully considered each point and have revised the manuscript accordingly. Below, we provide a point-by-point response to each comment. All changes to the manuscript are indicated with tracked changes. We believe these revisions have meaningfully strengthened the paper and we hope the revised version is suitable for publication in PLOS ONE.

The editor's formatting comments were addressed prior to this submission and are not repeated here. We respond below to the scientific comments from Reviewers 1 and 2.

Reviewer #1

Comment 1.1 — Chemoimmunotherapy Subgroup Analysis

Given the increasing adoption of neoadjuvant chemo-immunotherapy, a subgroup analysis focusing on patients treated with this strategy would be of particular interest, if feasible, even acknowledging the relatively limited sample size.

Response:

We thank the reviewer for raising this clinically important point. We agree that outcomes specifically within the chemoimmunotherapy subgroup are highly relevant given the widespread adoption of neoadjuvant nivolumab plus chemotherapy following CheckMate-816 and related trials.

We explored this analysis in detail. Of 2,257 total patients, only 146 (6.5%) received neoadjuvant chemoimmunotherapy, and critically, only 12 of these underwent sublobar resection (compared to 134 who received lobectomy). This imbalance reflects the real-world adoption timeline: chemoimmunotherapy was not widely used until after 2018, meaning the majority of our 2011–2021 cohort predates routine clinical use of checkpoint inhibitors in the neoadjuvant setting.

With only 12 sublobar patients in the chemoimmunotherapy group, a Kaplan-Meier or Cox regression analysis would be severely underpowered and statistically unreliable. Our institutional biostatistician confirmed that a formal survival comparison within this subgroup is not feasible with the current dataset. Importantly, the Cox model for the full stage III cohort already includes neoadjuvant treatment type as a covariate, and chemoimmunotherapy was independently associated with significantly improved survival compared to chemotherapy alone (HR=0.51, 95% CI 0.29–0.90, p<0.001), which highlights the prognostic importance of treatment regimen and supports the reviewer's interest in this subgroup.

We have added a sentence to the Limitations section acknowledging this constraint explicitly, and have strengthened the Future Directions section to highlight the chemoimmunotherapy era as a priority for future investigation.

Manuscript change:

Added to Limitations: 'The chemoimmunotherapy subgroup was similarly underpowered for subgroup survival analysis, with only 12 sublobar patients receiving this regimen.'

Added to Future Directions: emphasis on pCR-stratified analysis in the chemoimmunotherapy era as a priority for future prospective research.

Comment 1.2 — Pathological Upstaging/Downstaging

Did the authors evaluate patterns of pathological upstaging or downstaging following neoadjuvant therapy? An exploratory analysis of patients experiencing stage migration could provide additional insight into treatment response and surgical decision-making.

Response:

We thank the reviewer for this suggestion and explored it carefully. The NCDB does capture pathological stage, allowing us to construct a cross-tabulation of clinical versus pathological stage across the full cohort (n=2,257). Descriptively, among clinical stage III patients (n=1,943), pathological complete response as proxied by ypT0N0 (pathological stage 0) was observed only in 32 patients (1.4%), with partial downstaging to pathological stage I in 354 patients (15.7%) and stage II in 130 patients (5.8%). A total of 550 patients (24.4%) remained at pathological stage III with no evidence of downstaging.

However, a formal comparative analysis of downstaging rates between surgical groups was not feasible for two reasons. First, approximately 44% of clinical stage III patients (n= 854) had missing pathological stage data in the NCDB, a level of missingness that precludes reliable inference. Second, with only 148 sublobar patients in total (of whom a substantial proportion had missing pathological data), the sublobar subgroup within any pathological stage category was too small for meaningful comparison. Our institutional biostatistician also confirmed that any analysis based on these data would be underpowered and vulnerable to significant bias from missing data.

We have strengthened the Limitations section to explicitly acknowledge the missing pathological staging data as a constraint on this analysis.

Manuscript change: Added to Limitations: 'While pathological stage was available for a subset of patients, approximately 45% of stage III patients had missing pathological staging data, precluding reliable comparative analysis of downstaging rates between surgical groups.'

Comment 1.3 — Disease-Free Survival and Recurrence Data

While recognizing the limitations of the NCDB regarding recurrence data, further clarification on the lack of disease-free survival and recurrence information should be emphasized, as these outcomes would be particularly relevant when comparing sublobar resection and lobectomy in this setting.

Response:

We agree entirely with this comment. Disease-free survival and locoregional recurrence rate are arguably the most clinically meaningful endpoints when comparing extent of resection, as they directly assess adequacy of local tumor control, which is the central oncologic concern when considering sublobar resection. The reliance on overall survival as a primary endpoint is a recognized limitation of NCDB-based analyses in this surgical context, as it cannot disentangle locoregional failure from competing causes of death.

We have added explicit language to the Discussion to emphasize this point beyond the brief acknowledgment in the original Limitations section, and have strengthened the Limitations and Future Directions sections accordingly.

Manuscript change: Added to Discussion (Long-term outcomes section): 'Disease-free survival and locoregional recurrence are arguably the most clinically relevant endpoints when comparing extent of resection, as they directly capture the adequacy of local tumor control. The inability to assess these outcomes represents a meaningful constraint of NCDB-based analyses in this setting, and future studies should prioritize these endpoints over overall survival alone.'

Expanded in Limitations and Future Directions.

Reviewer #2

Major Comment — Pathological Response and pCR

An important element that appears to be missing from the analysis is the impact of pathological response to neoadjuvant therapy on survival outcomes. The neoadjuvant treatments included are highly heterogeneous (chemotherapy, chemoradiation, and chemo-immunotherapy) over a long study period. In the contemporary setting, neoadjuvant chemo-immunotherapy is associated with higher rates of major and complete pathological response, which are known to have strong prognostic value. It would therefore be relevant to assess whether outcomes differ according to pathological response (e.g. pCR or nodal downstaging), particularly within the sublobar resection cohort. If feasible within the NCDB, an analysis stratified by pathological response would significantly strengthen the manuscript. If not feasible, this limitation and its potential implications should be more explicitly discussed, acknowledging the prognostic role of pathological complete response and its possible interaction with surgical extent.

Response:

We thank the reviewer for this excellent and clinically important comment. We agree that pathological response, and pCR in particular, represents the variable most likely to identify patients in whom de-escalation of surgical extent might be oncologically appropriate. We explored this analysis in depth.

The NCDB does not capture pCR or major pathological response (MPR, defined as <10% residual viable tumor) as structured data elements. Pathological response scoring requires granular quantitative interpretation of narrative pathology reports, specifically, the percentage of residual viable tumor cells in the resected specimen, which falls outside the scope of standard cancer registry abstraction. This is a recognized and fundamental limitation of all national registry databases for research in the neoadjuvant therapy era.

The closest available proxy for pCR in the NCDB is pathological stage ypT0N0 (pathological stage 0), indicating no viable tumor in the specimen and no nodal disease. We examined this variable across the full cohort. Among clinical stage III patients (n=1,943), ypT0N0 was recorded in only 32 patients (1.4%). Given that sublobar resection represented only 6.4% of the overall cohort, we estimate fewer than 3 sublobar patients would fall into this pCR-proxy category, far too few for any meaningful survival analysis. Furthermore, approximately 45% of stage III patients had missing pathological stage data, further limiting the reliability of any response-stratified analysis.

MPR analysis is not feasible at all in this dataset, as percentage of residual viable tumor is not abstracted by cancer registrars and does not exist as a structured field in the NCDB.

We have substantially expanded the Discussion and Limitations sections to explicitly acknowledge these constraints, to describe what was and was not feasible with the available data, and to clearly articulate why pCR-stratified analysis represents a priority for future prospective research. We have also added language in the Future Directions section emphasizing that multi-institutional retrospective cohort studies using institutional tumor registries, where granular pathology data are accessible, represent the most feasible near-term approach to answering this question.

Manuscript change:

Added to Discussion (Role of sublobar resection section): New paragraph explicitly discussing pCR and MPR, explaining why ypT0N0-stratified analysis was not feasible (n=32 pCR proxies, <3 estimated sublobar, 45% missing path stage), and framing this as a priority for future prospective research.

Expanded Limitations section. Revised Future Directions section.

Minor Comment — Temporal Changes in Systemic Therapy

Given the long inclusion period (2011–2021) and the evolving neoadjuvant landscape, the potential impact of temporal changes in systemic therapy on outcomes could be further emphasized in the Discussion.

Response:

We agree with this comment and appreciate the opportunity to address it more explicitly. The 2011–2021 study period spans a substantial evolution in neoadjuvant treatment, from platinum-based chemotherapy as the dominant approach in the early years, to the integration of checkpoint inhibitors following landmark trials including CheckMate-816, NADIM, and NADIM-II in the later years.

Our multivariable Cox model adjusts for year of diagnosis as a covariate, which was independently associated with improved survival in later years (overall p=0.009 for year of diagnosis), consistent with the survival benefit conferred by improved systemic therapies over time. This variable serves as a partial proxy for evolving treatment paradigms. However, we acknowledge that year of diagnosis cannot fully account for the specific shift toward chemoimmunotherapy-based regimens after approximately 2018, nor the higher rates of pathological downstaging associated with these regimens.

We have added language to the Discussion to address this point, noting that the year-of-diagnosis covariate in our multivariable model provides partial adjustment for temporal treatment evolution while acknowledging its limitations as a proxy for regimen-specific effects.

Manuscript change:

Added to Discussion (Role of sublobar resection section): 'The evolving treatment landscape over the 2011–2021 study period, including the shift toward immunotherapy-based regimens after approximately 2018, is partially accounted for by year of diagnosis in our multivariable model (p=0.009), though this covariate cannot fully capture the differential pathological response rates associated with contemporary chemoimmunotherapy regimens.'

We hope these revisions adequately address the reviewers' concerns. We remain committed to transparency about the limitations of registry-based data in this evolving clinical landscape and believe the revised manuscript makes a meaningful contribution to the literature on surgical decision-making following neoadjuvant therapy in node-positive NSCLC.

We thank the editors and reviewers for their time and constructive engagement with our work.

Sincerely,

Feiran Lou, MD MS FACS

Division of Thoracic Surgery, UMass Chan Medical School

Feiran.lou@umassmemorial.org

---

## [Decision Letter · Decision Letter 1]

27 Apr 2026

Survival Following Sublobar Resection After Neoadjuvant Therapy for T1N1-2M0 Lung Cancer

PONE-D-26-01032R1

Dear Dr. Lou,

We’re pleased to inform you that your manuscript has been judged scientifically suitable for publication and will be formally accepted for publication once it meets all outstanding technical requirements.

Kind regards,

Luca Bertolaccini, M.D., Ph.D.

Academic Editor

PLOS One

Additional Editor Comments (optional):

Reviewers' comments:

Reviewer's Responses to Questions

**Comments to the Author**

1. If the authors have adequately addressed your comments raised in a previous round of review and you feel that this manuscript is now acceptable for publication, you may indicate that here to bypass the “Comments to the Author” section, enter your conflict of interest statement in the “Confidential to Editor” section, and submit your "Accept" recommendation.

Reviewer #1: All comments have been addressed

2. Is the manuscript technically sound, and do the data support the conclusions?

Reviewer #1: Yes

3. Has the statistical analysis been performed appropriately and rigorously? 

Reviewer #1: Yes

4. Have the authors made all data underlying the findings in their manuscript fully available?

Reviewer #1: Yes

5. Is the manuscript presented in an intelligible fashion and written in standard English?

Reviewer #1: Yes

6. Review Comments to the Author

Reviewer #1: The authors have addressed all my comments, and I believe the manuscript has improved accordingly. I have no further comments.

7. PLOS authors have the option to publish the peer review history of their article (what does this mean?). If published, this will include your full peer review and any attached files.

Reviewer #1: No

---

## [Editor Report · Acceptance letter]

PONE-D-26-01032R1

PLOS One

Dear Dr. Lou,

I'm pleased to inform you that your manuscript has been deemed suitable for publication in PLOS One. Congratulations! Your manuscript is now being handed over to our production team.

Kind regards,

on behalf of

Dr. Luca Bertolaccini

Academic Editor

PLOS One